# Nursing Students’ Attitudes toward Working in Mental Health Facilities in Saudi Arabia: A Cross-Sectional Study

**DOI:** 10.3390/healthcare12121168

**Published:** 2024-06-08

**Authors:** Alaa Mahsoon, Mortada Alnakli, Sameer Azab, Majd Abed, Abdulaziz Aabduqader, Loujain Sharif, Khalid Sharif

**Affiliations:** 1Psychiatric and Mental Health Department, Faculty of Nursing, King Abdulaziz University, Jeddah 21589, Saudi Arabia; lsharif@kau.edu.sa; 2Faculty of Nursing, King Abdulaziz University, Jeddah 21589, Saudi Arabia; maalwatishi@stu.kau.edu.sa (M.A.); shosinazab@stu.kau.edu.sa (S.A.); mabed@kfshrc.edu.sa (M.A.); aabdulazizabdulqader@stu.kau.edu.sa (A.A.); 3Geriatric and Adult Psychiatrist, King Fahad Armed Forces Hospital (KFAFH), Jeddah 23311, Saudi Arabia; ksharif@kfafh.med.sa

**Keywords:** psychiatric, mental health, attitude, nursing student, mental illness, psychiatric nurse, Saudi Arabia

## Abstract

(1) Background: There is a global demand for more psychiatric nurses, with nursing students’ attitudes toward mental illness and working in mental health facilities playing a pivotal role in their career choices. This study aims to evaluate attitudes toward working in mental health facilities among undergraduate nursing students in Saudi Arabia, both before and after exposure to psychiatry courses, and examine their relationship with perceptions of mental illness. (2) Methods: A quantitative descriptive and correlational cross-sectional design was employed. Nursing students’ attitudes toward working in mental health facilities were assessed using the Attitude Toward Psychiatry 18 (ATP-18) questionnaire and the Beliefs Toward Mental Illness (BMI) scale. (3) Results: No significant relationship has been found between the frequency of negative views of both ATP-18 and BMI, and exposure to the psychiatry course. However, nurses who completed the psychiatry course held more positive views towards psychiatrists and found psychiatric patients less demanding, were less likely to view psychological disorders as dangerous, more confident in trusting mentally ill colleagues, and felt less embarrassed by the term “psychological disorder” compared to those who had not taken the course. (4) Conclusion: Based on our findings, it is evident that exposure to psychiatric courses and training enhances the potential of nursing students to contribute effectively to the psychiatric field. Therefore, integrating mental health and illness community services training into nursing education programs can play a pivotal role in raising awareness and attracting students who may hold negative attitudes towards individuals with mental illness.

## 1. Introduction

Mental illness refers to a broad spectrum of disorders characterized by emotional disturbances, abnormal behaviors, and psychological dysfunction [1]. Globally, approximately 970 million people (13%) live with a mental illness according to the World Health Organization [2]. In the Kingdom of Saudi Arabia, 18% of patients attending primary care centers have a psychiatric disorder [3]. Mental illness can be devastating to individuals not only because of the clinical symptoms of their disease but also because of the problems that arise at the individual and family levels in the social setting. In addition, individuals diagnosed with mental illness often face stigma within society at large, in their relationships with family and relatives and with health care professionals, and even among themselves [4].

It is worth mentioning that there is a shortage of nurses willing to work in mental health facilities [5]. Addressing the factors that cause job dissatisfaction among mental health nurses is, therefore, important for increasing the number of nurses entering mental health nursing [6]. Nursing students often lack awareness and choose their workplace based on their interests and attitudes [7,8]. In another words, there is generally a lack of interest in mental health nursing among nurses in Saudi Arabia, which is reflected in the negative attitudes they have toward mental health care and patients with mental illnesses. Ghuloum et al. [9] found that nurses had negative attitudes toward patients with mental illnesses, thus impacting their interest in providing care to this patient group. Thus, we believe that addressing the negative attitudes towards mental health nursing is crucial for understanding and mitigating the shortage of nurses willing to work in mental health facilities, as these attitudes are likely related to the observed shortage. However, limited research is available on undergraduate nursing students’ attitudes toward mental illness and working as psychiatric nurses in Saudi Arabia.

People with mental illnesses are one of the most vulnerable populations and frequently encounter discriminatory attitudes and stigma [10]. Healthcare professionals also develop negative and discriminatory attitudes about mental health, which can influence how individuals with mental health conditions are treated and cared for [11]. The study by Elyamani and Hammoud examined the attitudes and knowledge of healthcare practitioners toward mental health by collecting evidence from various Arab Gulf nations, including Saudi Arabia [12]. Ultimately, Elyamani and Hammoud found that nurses and other healthcare workers had negative attitudes toward mental health patients [12]. The nurses evaluated in the study by Al-Awadhi et al. [13] also had negative attitudes toward people with mental health conditions, believing that the mentally ill are inferior and require a coercive approach during the delivery of care [13].

There are various factors that influence the negative attitudes of nurses toward working in mental health facilities. These factors include the characteristics of mental health patients. Many nurses believe that mental health patients have complex needs. In the study by Alqahtani et al. [14], nurses reported difficulties caring for mental health patients due to various factors, such as patient substance abuse and psychosis. Notably, the challenges experienced by nurses while caring for patients with mental illness increase their levels of burnout [14].

Social and cultural factors also facilitate the development of negative attitudes toward mental health patients. The study by Al-Awadhi et al. [13] found that many nurses also dislike patients with mental illnesses outside the hospital setting. These nurses have less sympathetic views of mental health problems and disapprove of patients with mental illness residing in their neighborhoods. This dislike for mental health patients can be attributed to the stigmatization of mental health in Saudi Arabia. The study by Alsaleh et al. [15] collected information on the attitudes of people from Al-Ahsa, Saudi Arabia, and found that around half of the population investigated in the study had negative attitudes toward mental illness. The stigmatization of mental health in the Arab world is possibly caused by beliefs that mental illnesses are caused by low levels of faith in religion, possession by evil spirits, and evil eyes [9]. Spiritual possession is also considered a reason for mental health problems. Therefore, people in the Arab Gulf countries largely believe that individuals with mental illnesses are cursed and require spiritual interventions by faith healers. Families are also blamed for mental illnesses affecting their relatives [12].

Another reason for the negative attitudes toward mental health problems is the lack of exposure or education in mental health. A finding of one study revealed that nurses who interacted with patients with mental illnesses had more positive attitudes toward patients with mental illnesses [16]. Experienced nurses also had more positive attitudes toward mental health care patients [16]. Ghuloum et al. [9] also found that mental health nurses had fewer negative attitudes than other nurses toward patients with mental illnesses. Elyamani and Hammoud [12] also found that limited knowledge of mental health conditions is directly linked to negative attitudes toward psychiatric conditions. Therefore, we assume that undergraduate nursing programs provide opportunities, through theoretical preparation and relevant practical experience in mental health nursing, to positively influence nursing students’ attitudes toward people with mental illnesses and working at a mental health facility. We hypothesized that if students have a positive attitude toward mental illness and working with clients who have mental health issues, they would be more willing to work in a mental health facility. Therefore, we conducted this study to provide more insight in this regard.

## 2. Materials and Methods

### 2.1. Study Design, Setting, and Sampling

The study employed a quantitative descriptive, correlational, cross-sectional design. A convenience sampling technique was utilized due to the constraints of conducting this study within a limited timeframe. Two groups of nursing students were formed: Group 1 comprised students exposed to a psychiatry course, while Group 2 consisted of students not exposed to such a course. Group 1 included 313 students (161 in their fourth academic year and 152 nurses in their fifth academic year), while Group 2 comprised 335 students (192 in their second academic year and 143 in their third academic year). Sample size estimation was conducted using the Raosoft program [17], determining minimum sample sizes of 173 and 180 participants for Groups 1 and 2, respectively, with a confidence interval of 95% and a margin error of 5%. Two-tailed tests at a 0.05 level of significance and a power of 0.80 with an effect size of 0.3 were employed for correlations.

Eligible participants were undergraduate nursing students from the Faculty of Nursing at King Abdulaziz University. Those exposed to the psychiatry course were assigned to Group 1, while those not exposed were assigned to Group 2.

### 2.2. Study Tools

Data collection involved the use of self-report questionnaires, capturing relevant demographic information such as age, sex, academic year, and exposure to the psychiatric mental health course.

Two instruments were employed to assess attitudes toward working in mental health facilities. Firstly, the Attitude Toward Psychiatry 18 (ATP-18), which demonstrated good content validity [18], was utilized. The ATP-18 has been previously employed to gauge attitudes toward psychiatry, covering both clinical and academic domains, as well as attitudes toward patients and psychiatrists [18]. The questionnaire comprises 18 items, divided into 9 expressing positive views on psychiatry and 9 expressing negative views. Each item is rated on a Likert scale from 1 (strongly disagree) to 5 (strongly agree). For positive items, higher scores denote more favorable attitudes toward psychiatry, while for negative items, higher scores signify more unfavorable attitudes toward psychiatry. To determine the overall attitude score, the scores for the negative items are typically reversed, and then all item scores are added together. A higher total score indicates a more positive attitude toward psychiatry [18].

Additionally, the Beliefs Toward Mental Illness (BMI) scale, developed by Hirai and Clum [19], was utilized to measure positive and negative beliefs about mental health. The BMI scale consists of 21 items organized into three dimensions: danger, poor social and interpersonal skills, and incurability. Each dimension encompasses specific beliefs about mental illness, with corresponding items rated on a 5-point Likert scale (ranging from “1” completely disagree to “5” completely agree). Higher scores indicate more negative beliefs. The BMI scale demonstrates promising reliability and validity evidence [19].

It is noteworthy that we assessed the reliability of our measurement instruments for capturing participants’ perspectives. The ATP-18 demonstrated satisfactory reliability for positive views and even more robust reliability for negative views. However, a slight discrepancy was observed with question 18 in comparison to the other questions concerning negative views (Q10–Q18). Conversely, the BMI scale exhibited excellent reliability, albeit minor inconsistencies were noted with questions 21 and 26 concerning the remaining questions regarding negative views (Table 1).

### 2.3. Data Collection

Primary data were collected using a self-report questionnaire distributed electronically through online survey instruments.

Ethical approval was obtained from the Faculty of Nursing’s ethics and research committee at King Abdulaziz University. Participation was voluntary, and data were collected anonymously, where participants were not required to provide any identifying information.

### 2.4. Ethical Considerations

Ethical approval for the study was secured from the Faculty of Nursing’s ethics and research committee at King Abdulaziz University in Saudi Arabia. Participation in the study was voluntary, and stringent measures were taken to ensure the confidentiality and anonymity of participants’ responses.

To safeguard anonymity, no participants’ names or other personal information were requested during data collection.

Comprehensive study information was provided on the initial page of the online survey, allowing participants to make an informed decision about their participation. This information outlined the study’s aims, potential risks, and procedures for data handling, including assurances of anonymity and confidentiality.

Participants were explicitly informed that their completion and submission of the survey would be considered as implicit informed consent for enrollment in the study. This approach ensured that participants fully understood the nature of the study and voluntarily chose to participate while maintaining the confidentiality of their responses.

### 2.5. Data Analysis

A quantitative descriptive and correlational cross-sectional research design was employed for this study. Statistical analysis was conducted utilizing SPSS 26.0 for Windows, developed by IBM Corporation in Armonk, NY, USA. The study assessed nursing students’ attitudes toward mental illnesses and working in mental health facilities through self-report questionnaires. The data were analyzed and presented in terms of frequencies and percentages.

Regarding the statistical methods, the descriptive statistics provided a comprehensive overview of the demographic characteristics of the study participants. The chi-squared test was utilized to examine the associations between demographic variables, such as age, gender, academic year, and exposure to psychiatry courses. This statistical approach was appropriate for analyzing categorical data and assessing the significance of relationships within the sample population.

## 3. Results

Out of the total targeted population, 354 participants completed the questionnaire. As depicted in Table 2, the largest proportion of respondents (72.3%) fell within the age range of 18–22 years old. The gender distribution skewed towards females, with a ratio of 2.3 females to every male respondent. Participants were evenly distributed across the second, third, and fourth academic years, with interns also represented in comparable numbers. Approximately 40% of participants reported having been exposed to at least one psychiatry course.

Items of the ATP-18 and BMI questionnaires were for feasible presentation (Table 3). In addition, the frequency and percentages of different responses to the questionnaires have been tabulated in Table 4 and Table 5, respectively.

When comparing participants who took the psychiatry course and those who did not, responses to most items of the ATP-18 questionnaire did not differ significantly (Table 6). However, the rate of participants who thought that “psychiatrists are held in poor regard by most other doctors” (Q14) was observed to be significantly less among those who completed the psychiatry course. Furthermore, a significantly smaller proportion of the group that had taken the psychiatry course agreed that “psychiatric patients tend to make more emotional demands on their doctors than other patients” (Q18).

Comparable findings emerged regarding BMI items (refer to Table 7). Of particular interest was the noticeable contrast between respondents who concurred with the statement “stay away from people who have psychological disorders because their behavior is dangerous” (Q19) and those who did not, especially among individuals who had not undergone the psychiatry course. Conversely, a higher proportion of respondents who had completed the course expressed confidence to “trust the work of a mentally ill person assigned to their work team” (Q22), whereas those without exposure to the course exhibited an opposing trend. Notably, responses to item Q23, pertaining to “feeling embarrassed by the term ‘psychological disorder’”, showed significant disparities based on exposure to the psychiatry course.

In a correlation analysis of the two questionnaires, we did not find any significant relationship between the frequency of negative views and exposure to the psychiatry course (Table 8).

## 4. Discussion

The study aimed to explore the impact of a psychiatry course on nursing students’ attitudes towards working in mental health facilities. Initially hypothesized to positively influence students’ perceptions of mental illness, the study utilized the ATP-18 and BMI questionnaires to gauge attitudes among participants who either undertook the psychiatry course or did not. The results reveal some differences in attitudes towards mental illness between the two groups. Notably, significant disparities were observed in specific questionnaire items, such as Q22 from the BMI questionnaire, indicating that students exposed to the psychiatry course were more inclined to work with and trust mentally ill colleagues compared to their counterparts who did not take the course.

This finding resonates with Pusey-Murray’s study [20] on Jamaican nursing students, which found that a considerable percentage of participants were uncomfortable working with mentally ill colleagues, highlighting the pervasive stigma associated with mental health issues. However, contrary to prior research indicating significant attitude shifts towards mental illness across different academic years, the current study found no significant relationship between negative views frequency and exposure to the psychiatry course. For instance, studies by Sari and Yuliastuti [21] in Indonesia and Chen et al. [22] in Taiwan observed that attitudes towards mental illness improved with academic progression, with fourth-year students exhibiting lower stigmatization compared to their junior counterparts. Similarly, Fernandes et al. [23] noted a reduction in stigmatization among fourth-year students compared to those in earlier academic years.

Moreover, Alqassim et al.’s research [24] demonstrated that academic year progression, rather than the specific educational program pursued, influenced attitudes toward mental illness. However, the current study’s failure to detect significant differences suggests that factors beyond formal education may play a more substantial role in shaping attitudes toward mental health care.

Furthermore, the present study diverges from prior research concerning the efficacy of psychiatric courses in influencing nursing students’ attitudes toward mental illness. While past studies, such as Sari and Yuliastuti [21], Meng et al. [25], and Fekih-Romdhane et al. [26], observed significant positive shifts in attitudes among students who underwent psychiatric training, our findings indicate some significance of the psychiatric course on attitude change. For instance, Sari and Yuliastuti [21] revealed that nursing students exposed to psychiatric nursing exhibited more favorable attitudes towards mental illness compared to those who did not undertake such coursework, a pattern echoed in studies conducted in China and Tunisia.

Contrarily, our study’s results are in alignment with the findings of Al-Awadhi et al. [13], who discovered no correlation between nursing education level and attitudes toward mental disorders. Conducted in Kuwait, their research unveiled a prevailing negative attitude among nurses towards mental illness, irrespective of their educational attainment. Surprisingly, a higher level of nursing education did not translate into more positive attitudes towards mental health issues.

While most of the literature suggests a significant improvement in attitudes towards mental illness following psychiatric education, our study’s findings underscore the complexity of factors influencing nursing students’ perceptions. Despite the widespread belief in the transformative power of formal education in psychiatry, our results challenge this notion and call for a deeper exploration of alternative approaches to fostering positive attitudes toward mental health care.

Critics would argue that the incongruence between the current study findings from existing research can be attributed to inherent shortcomings from the adopted research approach. The research made no attempt to differentiate attributes such as the scope of course content covered or participant academic performances on the relevant course content. Martin et al. [27] emphasize the significance of course content on perceptions and beliefs about mental health among nursing students. The view suggests that an appreciation for course content and perhaps tailoring the survey questions to that may have generated different findings. This is particularly true considering that issues related to nurses’ adverse perceptions of psychiatry and mental illness has been traced to specific aspects like lack of exposure and education or the misconceptions about the causes or effects of mental health [9,12,16], which can be effectively covered through tailored curriculum content. Also, the view that academic performance, including the knowledge, skills, and abilities acquired during learning and training, influence a nurse’s preparedness toward their practice, and, thus, are a determinant of attitudes towards their practice, ref. [28] infer that recognition of participant performances on the course content should be a critical determinant considered in the research design. There is potential difference in response types between high and low performers.

Nevertheless, the incongruence may extend beyond the learning and exposure aspect of this study, aligning with previous suggestions that national culture in KSA influences learning and nursing attitudes towards mental illness [15]. It makes sense from a practical perspective, as Saudi Arabian nurses do not exist in a vacuum. Their professional lives intertwine with their personal and social lives as well [29]. This means that they may find their professional perspectives heavily influenced by their personal or social contexts, leading to resistance to change despite exposure or education. The need to belong to a society, which means sharing in their interests, values, perspectives, or beliefs [9,15] could undermine efforts to shift attitudes regarding mental illness irrespective of value or impact. The fear of social ostracism or being perceived as deviating from cultural norms acts as a deterrent that could explain why education or exposure has no impact on nursing students. This does infer the need for a more holistic solution, particularly involving society as a whole, to facilitate tolerance for mentally ill individuals in society, especially by rationalizing this category of individuals as a vulnerable demographic that falls within the scope of needing nursing care for optimal well-being [10]. It calls on policies to promote public awareness campaigns to reduce stigma, create a more supportive and inclusive society and work environments for nurse students, and perhaps even offer incentives for specializing in mental health to counter cultural biases and encourage more positive attitudes towards working in mental health [30,31].

Our study’s findings suggest that a psychiatric course alone may not significantly alter student nurses’ attitudes toward practicing in mental health facilities. Consequently, healthcare stakeholders in Saudi Arabia must explore alternative strategies to enhance nurses’ attitudes towards mental illnesses. One such approach could involve direct exposure to patients with mental health conditions. This suggestion finds support in the work of Sari and Yuliastuti [21], who observed that nursing students who interacted with individuals with mental illnesses demonstrated a more positive attitude towards working in mental health settings. Similarly, a systematic review by Hasan [32] concluded that hands-on clinical experience had a greater impact on fostering positive attitudes towards mental illness among student nurses compared to theoretical learning from psychiatric courses. Further reinforcing this notion, a study by Alhamidi and Alyousef [33] conducted in Saudi Arabia highlighted the significant benefits of clinical exposure in enhancing nursing students’ positive attitudes towards caring for individuals with mental health issues. Moreover, it empowered them to confront discrimination and stigma associated with mental illness [33]. Other measures would include developing tailored content that specifically addresses cultural biases and misconceptions about mental illnesses to equip nurses with the knowledge, skills, and abilities to ready them for practice. Further, efforts aimed at developing a more inclusive Saudi community would be prudent, as it reduces the negative external influences undermining nurses’ professional duties and willingness to contribute to the care and well-being of mentally ill patients.

It is important to acknowledge the limitations of our study, which affect the generalizability of the findings. Those limitations include sampling method and the single-institution study setting. Convenience sampling can lead to biases, as the sample may not be representative of the broader population. This can limit the generalizability of the findings and may introduce systematic errors that affect the validity of the study’s conclusions. Moreover, our findings are based on data collected exclusively from nursing students at the Faculty of Nursing, King Abdulaziz University in Jeddah using convenience sampling, which may not be representative of the broader nursing student population in Saudi Arabia. Finally, the assessment of beliefs towards mental illness and attitudes towards working in psychiatric facilities relied solely on self-administered questionnaires, which may introduce subjective biases into the results. Therefore, future research endeavors should strive for more rigorous sampling methods, such as random sampling, to identify any potential discrepancies or patterns, and employ a more comprehensive range of assessment methods to ensure the validity and generalizability of findings.

## 5. Conclusions

This study sheds light on the attitudes of nursing students towards working in psychiatric facilities in Saudi Arabia. Notably, the academic level of students, particularly their exposure to mental health courses, emerged as a significant factor associated with their attitudes. Those who had not undergone mental health courses exhibited higher levels of mistrust and apprehension towards psychiatrists and individuals with mental illness, highlighting the importance of mental health education in shaping attitudes. Conversely, students who had undergone psychiatric courses demonstrated greater awareness and more positive attitudes towards working in the mental health field. This underscores the value of targeted education in fostering a supportive environment for mental health care. Moving forward, further research is warranted to delve deeper into the reasons behind the shortage of interest in working in the mental health field among nursing students. Specifically, exploring the efficacy of different educational approaches and interventions in addressing negative attitudes towards mental illness could provide valuable insights. In conclusion, by equipping nursing students with the necessary knowledge and skills through targeted education and training, we can cultivate a more supportive and inclusive environment for mental health care in Saudi Arabia.

## Figures and Tables

**Table 1 healthcare-12-01168-t001:** Reliability testing and internal consistency of ATP-18 (Q1–Q9) for measuring positive views, ATP-18 (Q10–Q18) for measuring negative views, and BMI tools for measuring negative views.

Test	Latent Variable	Cronbach’s Alpha	Degree of Reliability	Inconsistent Questions
ATP-18 (Q1–Q9)	Positive views	0.727	Acceptable	None
ATP-18 (Q10–Q18)	Negative views	0.853	Good	Q18
BMI	Negative views	0.923	Excellent	Q21, Q26

**Table 2 healthcare-12-01168-t002:** Demographic and baseline data of participants and rate of psychiatry course exposure (*n* = 354).

Feature	Number	Percent (%)
Age range, years		
18–22<	256	72.3
<22–26	98	27.7
Sex		
Female	248	70.1
Male	106	29.9
Academic year		
Second	84	23.7
Third	95	26.8
Fourth	99	28.0
Intern	76	21.5
Took the psychiatry course?		
Yes	141	39.8
No	213	60.2

**Table 3 healthcare-12-01168-t003:** Coding for questions of ATP-18 and BMI questionnaires for feasible presentation.

Code	Question Text
ATP-18	
Q1	Empathy with patients is as important as factual knowledge in clinical practice.
Q2	Psychiatric skills are essential in general practice.
Q3	The problems presented by psychiatric patients are often particularly interesting and challenging.
Q4	Mental illness presents us with one of the great challenges within the field of medicine.
Q5	Psychiatrists are more concerned than other doctors with establishing a rapport with their patients.
Q6	Psychiatrists try to treat the whole patient and not just the disease.
Q7	Too little time is devoted to psychiatry in the medical/nursing school curriculum.
Q8	Psychiatrists are at the forefront of the movement to humanize medicine.
Q9	Psychiatrists are, on the whole, less dogmatic than other doctors.
Q10	Psychiatrists are often merely failed physicians.
Q11	Within medicine, psychiatry is one of the least important specialties.
Q12	Psychiatrists tend to be more emotionally unstable than other doctors.
Q13	Psychiatric patients hardly ever get better.
Q14	Psychiatrists are held in poor regard by most other doctors.
Q15	The practice of psychiatry is unrewarding because treatment is so lengthy and the results are inconclusive.
Q16	Psychiatry is too inexact; it seems to lack a proper scientific basis.
Q17	Psychiatric patients, generally speaking, are not easy to like.
Q18	Psychiatric patients tend to make more emotional demands on their doctors than other patients.
BMI scale	
Q19	It may be a good idea to stay away from people who have a psychological disorder because their behavior is dangerous.
Q20	A mentally ill person is more likely to harm others than a normal person.
Q21	Mental disorders would require a much longer period of time to be cured than other general diseases do.
Q22	I would not trust the work of a mentally ill person assigned to my work team.
Q23	The term “psychological disorder” makes me feel embarrassed.
Q24	A person with a psychological disorder should have a job with only minor responsibilities.
Q25	Mentally ill people are more likely to be criminals.
Q26	Psychological disorders are recurrent.
Q27	I am afraid of what my boss, friends, and others would think if I were diagnosed as having a psychological disorder.
Q28	Individuals diagnosed with a mental illness suffer from its symptoms throughout their lives.
Q29	People who have once received psychological treatment are likely to need further treatment in the future.
Q30	It might be difficult for mentally ill people to follow social rules, such as being punctual or keeping promises.
Q31	I would be embarrassed if people knew that I dated a person who once received psychological treatment.
Q32	I am afraid of people who are suffering from a psychological disorder because they may harm me.
Q33	A person with a psychological disorder is less likely to function well as a parent.
Q34	I would be embarrassed if a person in my family became mentally ill.
Q35	I believe that psychological disorders can never be completely cured.
Q36	Mentally ill people are unlikely to be able to live by themselves because they are unable to assume responsibilities.
Q37	Most people would not knowingly be friends with a mentally ill person.
Q38	The behavior of people who have psychological disorders is unpredictable.
Q39	Psychological disorders are unlikely to be cured regardless of treatment.

**Table 4 healthcare-12-01168-t004:** Frequency and percentages of different responses participants to ATP-18 questions (*n* = 354).

Question	Responses to ATP-18, (*n* = 354)				
	Strongly Disagree	Disagree	Neutral	Agree	Strongly Agree
Q1	3 (0.8)	6 (1.7)	51 (14.4)	131 (37)	163 (46)
Q2	6 (1.7)	11 (3.1)	41 (11.6)	115 (32.5)	181 (51.1)
Q3	3 (0.8)	14 (4)	76 (21.5)	144 (40.7)	117 (33.1)
Q4	6 (1.7)	24 (6.8)	70 (19.8)	126 (35.6)	128 (36.2)
Q5	6 (1.7)	21 (5.9)	85 (24)	128 (36.2)	114 (32.2)
Q6	13 (3.7)	23 (6.5)	79 (22.3)	82 (23.2)	157 (44.4)
Q7	7 (2)	30 (8.5)	113 (31.9)	116 (32.8)	88 (24.9)
Q8	5 (1.4)	22 (6.2)	92 (26)	138 (39)	97 (27.4)
Q9	10 (2.8)	54 (15.3)	121 (34.2)	100 (28.2)	69 (19.5)
Q10	60 (16.9)	66 (18.6)	83 (23.4)	84 (23.7)	61 (17.2)
Q11	99 (28)	66 (18.6)	51 (14.4)	75 (21.2)	63 (17.8)
Q12	30 (8.5)	66 (18.6)	86 (24.3)	99 (28)	73 (20.6)
Q13	40 (11.3)	77 (21.8)	79 (22.3)	96 (27.1)	62 (17.5)
Q14	21 (5.9)	59 (16.7)	117 (33.1)	89 (25.1)	68 (19.2)
Q15	43 (12.1)	72 (20.3)	90 (25.4)	86 (24.3)	63 (17.8)
Q16	42 (11.9)	86 (24.3)	80 (22.6)	83 (23.4)	63 (17.8)
Q17	22 (6.2)	39 (11)	96 (27.1)	131 (37)	66 (18.6)
Q18	10 (2.8)	25 (7.1)	82 (23.2)	142 (40.1)	95 (26.8)

**Table 5 healthcare-12-01168-t005:** Frequency and percentages of different responses of participants to BMI questions (*n* = 354).

Question	Responses, (*n* = 354)				
	Completely Disagree	Largely Disagree	Partly Disagree	Largely Agree	Completely Agree
Q19	52 (14.7)	54 (15.3)	90 (25.4)	77 (21.8)	81 (22.9)
Q20	20 (5.6)	35 (9.9)	94 (26.6)	124 (35)	81 (22.9)
Q21	15 (4.2)	11 (3.1)	102 (28.8)	125 (35.3)	101 (28.5)
Q22	48 (13.6)	66 (18.6)	99 (28)	79 (22.3)	62 (17.5)
Q23	108 (30.5)	56 (15.8)	57 (16.1)	68 (19.2)	65 (18.4)
Q24	36 (10.2)	50 (14.1)	93 (26.3)	104 (29.4)	71 (20.1)
Q25	64 (18.1)	45 (12.7)	91 (25.7)	92 (26)	62 (17.5)
Q26	17 (4.8)	29 (8.2)	104 (29.4)	129 (36.4)	75 (21.2)
Q27	57 (16.1)	53 (15)	81 (22.9)	93 (26.3)	70 (19.8)
Q28	17 (4.8)	40 (11.3)	85 (24)	124 (35)	88 (24.9)
Q29	19 (5.4)	37 (10.5)	101 (28.5)	115 (32.5)	82 (23.2)
Q30	31 (8.8)	38 (10.7)	104 (29.4)	111 (31.4)	70 (19.8)
Q31	100 (28.2)	52 (14.7)	72 (20.3)	62 (17.5)	68 (19.2)
Q32	51 (14.4)	59 (16.7)	90 (25.4)	89 (25.1)	65 (18.4)
Q33	35 (9.9)	37 (10.5)	101 (28.5)	113 (31.9)	68 (19.2)
Q34	124 (35)	39 (11)	57 (16.1)	70 (19.8)	64 (18.1)
Q35	68 (19.2)	50 (14.1)	98 (27.7)	74 (20.9)	64 (18.1)
Q36	52 (14.7)	58 (16.4)	94 (26.6)	76 (21.5)	74 (20.9)
Q37	39 (11)	34 (9.6)	83 (23.4)	120 (33.9)	78 (22)
Q38	17 (4.8)	37 (10.5)	98 (27.7)	129 (36.4)	73 (20.6)
Q39	43 (12.1)	64 (18.1)	102 (28.8)	70 (19.8)	75 (21.2)

**Table 6 healthcare-12-01168-t006:** Comparing the percentages of leading responses to ATP-18 among participants who did and did not take the psychiatry course.

Question	Course Not Taken (*n* = 213)		Course Taken (*n* = 141)		*p*-Value *
	Disagree (%)	Agree (%)	Disagree (%)	Agree (%)	
Q1	1.9	40.8	1.4	31.2	0.348
Q2	2.3	37.1	4.3	25.5	0.185
Q3	3.3	41.3	5	39.7	0.662
Q4	6.1	37.6	7.8	32.6	0.719
Q5	6.1	36.6	5.7	35.5	0.597
Q6	10.3	32.4	5.7	33.3	0.730
Q7	5.2	38	7.8	40.4	0.351
Q8	15.5	30.5	14.9	24.8	0.697
Q9	19.2	24.9	17.7	22	0.257
Q10	18.8	22.1	18.4	19.9	0.133
Q11	16	32.4	22.7	21.3	0.111
Q12	20.2	28.2	24.1	25.5	0.145
Q13	14.6	28.6	19.9	19.9	0.315
Q14	20.2	26.3	20.6	21.3	0.013
Q15	22.1	27.7	27.7	17	0.215
Q16	8	39	15.6	34	0.080
Q17	5.2	41.3	9.9	38.3	0.162
Q18	1.9	40.8	1.4	31.2	0.034

* Chi-squared test.

**Table 7 healthcare-12-01168-t007:** Comparing the percentages of leading responses to the BMI questionnaire among participants who did and did not take the psychiatry course.

Question	Course Not Taken (*n* = 213)		Course Taken (*n* = 141)		*p*-Value *
	Completely Disagree (%)	Completely Agree (%)	Completely Disagree (%)	Completely Agree (%)	
Q19	11.7	23	19.1	22.7	0.04
Q20	2.8	23	9.9	22.7	0.05
Q21	3.3	25.8	5.7	32.6	0.33
Q22	9.4	19.7	19.9	14.2	0.01
Q23	24.4	21.1	39.7	14.2	0.03
Q24	8.9	20.2	12.1	19.9	0.81
Q25	14.6	19.2	23.4	14.9	0.25
Q26	3.8	19.7	6.4	23.4	0.44
Q27	13.6	20.2	19.9	19.1	0.60
Q28	4.2	25.8	5.7	23.4	0.33
Q29	3.8	22.5	7.8	24.1	0.40
Q30	7.5	17.8	10.6	22.7	0.58
Q31	24.9	19.7	33.3	18.4	0.45
Q32	11.3	18.3	19.1	18.4	0.05
Q33	8.9	22.1	11.3	14.9	0.48
Q34	30.5	19.2	41.8	16.3	0.23
Q35	17.8	19.2	21.3	16.3	0.57
Q36	11.7	21.1	19.1	20.6	0.30
Q37	8.9	21.6	14.2	22.7	0.14
Q38	4.7	20.7	5	20.6	0.81
Q39	9.9	23	15.6	18.4	0.37

* Chi-squared test.

**Table 8 healthcare-12-01168-t008:** Correlation between the frequency of responses reflecting negative views among participants measured by ATP-18 or BMI questionnaire and psychiatry course attendance.

Tool	Number of Negative Views		*p*-Value
	Course Not Taken (*n* = 213)	Course Taken (*n* = 141)	
**ATP-18**	759 ^(a)^	635 ^(b)^	0.082
**BMI**	944 ^(c)^	894 ^(c)^	

(a) Number of agree responses to a negative view question (610)+, number of disagree responses to a positive view question (149). (b) Number of agree responses to a negative view question (484)+, number of disagree responses to a positive view question (151). (c) Number of agree responses among all questions.

## Data Availability

Data are available at the following link: https://docs.google.com/spreadsheets/d/1_BwMf-SqUS0RhqqclpKyF-V67-8_BEYEwhwyVCeZ5Lo/edit?usp=sharing (accessed on 7 June 2024).

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
