# Peer review of "Nursing Students’ Attitudes toward Working in Mental Health Facilities in Saudi Arabia: A Cross-Sectional Study"

_healthcare, 2024, doi:10.3390/healthcare12121168_

Round 1

Reviewer 1 Report

Comments and Suggestions for Authors

Dear Authors

In reviewing the manuscript titled "Nursing Students’ Attitudes Toward Working in Mental Health Facilities in Saudi Arabia," several key concerns need attention. Please find my comments below:

Abstract

Ensure that the abstract clearly outlines the significance of the findings and their implications for practice and policy. It should summarise the problem, methods, significant results, and conclusions. Enhance the last sentence to state the findings’ practical implications and potential impact explicitly.

Introduction

Expand on the global context, comparing it with regional specifics in Saudi Arabia to highlight the study's unique aspects. Emphasise the gap in the literature that your study addresses.

Strengthen the rationale by linking to specific health outcomes, policy implications, or educational reforms that underscore the urgency and relevance of the research.

Methods

Provide more detailed descriptions of the data collection instruments, including validation studies and cultural adaptations for the Saudi context.

The article mentions the use of convenience sampling, which could limit the generalizability of the findings. Specifically, "A convenience sampling technique was used" without sufficient justification for why this method was chosen over more robust sampling methods. Discuss the choice of convenience sampling more critically, including its limitations and how it impacts the generalizability of the results.

 While the article mentions obtaining ethical approval and informed consent, it lacks a detailed discussion on how participants were informed about the study's aims, potential risks, and the confidentiality of their responses. For instance, "Enrollment in the study was optional, and all data were collected in an entirely anonymous manner" lacks detail on ensuring anonymity and managing sensitive data.

Results

Provide a better explanation of the statistical methods and why they were appropriate for your study design. Discuss any adjustments made for potential confounders.

Discussion

The study's contribution to existing knowledge, particularly how it extends or challenges the current understanding of nursing attitudes towards mental health in Saudi Arabia, is unclear. The manuscript should highlight unique insights or implications that could quickly review existing practices or theories in mental health education:

-       Strengthen the connection between your results and existing research. Highlight where your findings align with or differ from previous studies and hypothesise why.

-       Elaborate on how these findings could influence nursing education curricula, policy-making, or clinical practices, specifically in mental health settings.

-       Provide a more comprehensive critique of the study's limitations, including how they might be addressed in future research.

Conclusions

Give specific, evidence-based recommendations that can be implemented by policymakers, educators, or clinical managers.

Suggest areas where further studies are needed, suggesting potential methodologies or particular aspects of attitude changes requiring more research.

Comments on the Quality of English Language

Ensure the language is formal, precise, and appropriate for an academic audience. Avoid repetitive phrases and keep the writing concise.

Author Response

Thank you for your comments. please note that all your comments have been considered and the manuscript edited accordingly. Please see the attachment." i

Reviewer 2 Report

Comments and Suggestions for Authors

Thank you for addressing an important topic related to mental health nursing.

There are major notes that could be modified/improved before we move forward in this manuscript.  

Abstract:

1) Actually there are no results. you presented a method rather than a result. 

2) Your conclusion should be in align with the results.

Introduction

3) A major concern is; how do measuring attitude toward psychiatry and believes toward mental illness was considered an attitude toward working in mental health? Even, you did not ask the student if they thinking to work in psychiatric setting or not. Therefore, I think you need to make this clear and meaningful to the readers. 

4) Be consistent in using the word "Arabian Gulf", in some places you have only Gulf. 

5) Alshowkan , line 75, are not found on the reference list. reference number 10 is Tay et al, 2004. Please cross-reference between references in the text and in the list.

6) In line 95, reference number 5 is for Alyammani bot for Ghuloum. Same thing for Hasan line 294. 

7) Lines 103-106 are repetition of the previous sentence.  

Study design, population and sample

8) Line 113 you are talking about graduates. This is confusing for readers. Graduate students are those studying in master and phd, however your sample from nursing students who are doing their internship; which is literally the 5th year of Bachelor educational degree. 

Study tool

9) We need validity and reliability data about ATP-18, as well as the scoring system. 

10) There discrepancies in using the words of disagree, mostly dis-agree ,,,,Line 146-148). firstly there are no 0 as I noticed in table 4. Also, scores of 0-5 is a 6-point likert-type scale. Further, you have different words in the table 4 to represent those scores (0-6). Be clear and consistent with that in the text and in table 4. 

Data analysis

11) please indicate why you used chi-squared?

12) I did not see in the results any thing related to logistic regression analysis. 

Results

13) Table 1, be  aware of exclusive and inclusive in using the number 22 for age. 

14) It is not clear of how you calculated % of Disagree and % of Agree in table 5.

15) It is not clear of how you calculated % of Comm. Disagree and % of Com. Agree in table 6. and what is Com.? 

16) Table 7 is completely confusing. what statistic you used? how did you calculate number of negative views? number of Agree and disagree? 

Discussion

17) in line 240, and in line 271, indicate what do you mean by minimal? 

18) line 247, no start with numbers.

19)  The major point in this section is; you did not discuss why your result is incongruent with national and international results. In other word, why Saudi Arabian nursing students did not change their attitude toward mental illness after being exposed to a psychiatric course. You may describe earlier the psychiatric course taken (theory? clinical? both?)

Reference: 

20) Check again and follow the Journal style. Good Luck 

Author Response

(The authors gave the same response as above.)

Round 2

Reviewer 1 Report

Comments and Suggestions for Authors

Dear Authors. Thank you very much for your dedicated efforts in addressing the feedback provided and substantially improving the article.  Further comments below:

While convenience sampling has been justified, a more critical discussion of its limitations is needed.

There is still the need to contextualize the findings within the broader research landscape.

While the revisions have addressed some of the study's limitations, a more comprehensive critique could be beneficial. This includes discussing potential biases introduced by the sampling method and the single-institution study setting, which could affect the generalizability of the findings.

Comments on the Quality of English Language

Minor editing of English language required

Author Response

Thank you for your valuable feedback and for recognizing our efforts to improve the manuscript. We appreciate your further comments and have made the following revisions to address them comprehensively:

Reviewer 2 Report

Comments and Suggestions for Authors

Thank you for revising the manuscript. It is much better in this version. However, few minor notes need to be fixed.

1.      In the first revision, my comment was “A major concern is; how do measuring attitude toward psychiatry and believes toward mental illness was considered an attitude toward working in mental health? Even, you did not ask the student if they thinking to work in psychiatric setting or not. Therefore, I think you need to make this clear and meaningful to the readers”. I read your clarification in the authors’’ responses. I want to see this clarification in the introduction part to be read by the readers. Also, support your explanation that “We have noticed a shortage of nurses willing to work in mental health facilities. Additionally, nursing students often lack awareness and choose their workplace based on their interests and attitudes. We hypothesized that if students have a positive attitude toward mental illness and working with clients who have mental health issues, they would be more willing to work in a mental health facility. Therefore, we conducted this study to provide more insight in this regard” with suitable reference.

2.      Table 2, be aware of exclusive and inclusive in using the number 22 for age. I mean you cannot have 22 in both categories. In other words, students with 22 years old; are they in first group or second. Even if the tool used it in this way, you need to change.

3.      In table 8, I would use a, b, and c instead of *, **, and *** (as these refers mainly to significance).

44.      References: some references need to be modified to conform to the journal style. E.g full name of journals vs abbreviated names.

55.      The authors refer the reader to many appendices. Do the readers able to view those appendices?  

Author Response

(The authors gave the same response as above.)
